# Health-Related Quality of Life in Patients with Health Conditions in Lebanese Community Setting

**DOI:** 10.3390/ijerph18168817

**Published:** 2021-08-21

**Authors:** Hani Dimassi, Soumana C. Nasser, Aline Issa, Sarine S. Adrian, Bassima Hazimeh

**Affiliations:** 1School of Pharmacy, Lebanese American University, Byblos 1401, Lebanon; Hani.Dimassi@lau.edu.lb (H.D.); sarine.aderian@gmail.com (S.S.A.); 2School of Pharmacy, Saint Joseph University, Beirut 1100, Lebanon; aline.issa@icloud.com; 3Institute of Public Health, Saint Joseph University, Beirut 1100, Lebanon; bassima.hazimeh@net.usj.edu.lb

**Keywords:** HRQoL, EQ-5D-5L, utility measures, health economics, cardiovascular, diabetes

## Abstract

Background: The measurement of health-related quality of life (HRQoL) provides utility scores that could be used for health economics assessment. The aim of this study was to measure HRQoL in Lebanese patients with certain medical conditions, and to determine demographic and medical factors affecting such health utility scores. Method: This was a prospective cross-sectional pilot study conducted to gather information on the socioeconomic status, health condition and quality of life of participants with common diseases during their community pharmacy visit. The EuroQol-5-Dimension instrument was used to measure utility scores and SPSS v26 was used to perform the statistical analysis. Results: Participants (*n* = 102) gave an average of 6.8 and 7.4 out of 10 for their current health and for their satisfaction with their treatment, respectively. The mean utility score was 0.762 (SD 0.202). The number of prescribed medications per respondent indicated a significant impact on HRQoL (*p* = 0.002). On average, the utility scores were low for participants who were 75 years or older (0.15, *p* < 0.001), and those who were hospitalized in the past 12 months (0.111, *p* < 0.001). For every unit increase in treatment satisfaction, the quality-of-life score increased by 0.036 unit (*p* = 0.001). Conclusion: This pilot study measured health utility scores and factors influencing HRQoL in the Lebanese population. Further studies are needed to confirm our findings and to develop and validate tools helping to measure health related quality of life in the population in Lebanon.

## 1. Introduction

In recent years, the competitive global market place has increasingly acknowledged the usefulness of health-related quality of life (HRQoL) data to assist policy makers in reimbursement decisions by identifying the best alternative option among innovative medicines [1]. EuroQol 5 Dimension (EQ-5D-5L) is an instrument used to evaluate the generic quality of life developed in Europe and is a preference-based HRQoL measure. This descriptive instrument has five questions each addressing one of the following five dimensions: mobility, self-care, usual activities, pain/discomfort, and anxiety/depression, which are common problems in patients with chronic diseases [2].

The impact of common conditions on health status and their symptomatic management are consistently ranked as the highest priorities for patients and healthcare providers [2,3]. Furthermore, there is a growing emphasis on engaging the patient in his/her own care to optimize patient outcomes, and justifying whether or not the additional cost is worth the additional effectiveness of a new treatment option compared to the standard option [1]. Thus, policy-makers worldwide are supporting the decisions of providing patient access to innovative therapy based on cost effectiveness analysis using health outcomes that measure patient preference or HRQoL in addition to clinical surrogate endpoints [4]. In Lebanon, there is an increased need for HRQoL data to conduct health economics analysis and due to the negative or positive influence of social determinants on health, such as employment, education, access to health services. A recent study conducted in adults in Jordan and Lebanon reported a negative impact of diabetes mellitus on patients’ quality of life and satisfaction with their treatment [5]. 

The increasing epidemiologic and economic burden of highly prevalent diseases such as cardiovascular disease, diabetes mellitus and non-communicable diseases warrants attention [6]. 

The Lebanese healthcare sectors have been struggling with healthcare budget constraints, and facing challenges to support their reimbursement decisions with evidence data on the value of interventions, cost effectiveness evaluation, managed entry agreement schemes, as well as HRQoL. Therefore, the importance of cost effectiveness analysis or cost utility analysis has been recognized and decision makers are seeking information and tools that can assist in reimbursement decisions and assessment of interventions [6,7]. While HRQoL measures are crucial for determining population-specific QALYs, country-specific EQ-5D-5L value set and health preference measures are missing in Lebanon. In order to support the outcomes of interventions, Lebanese policy-makers rely on the results of clinical trials and/or data extrapolated from CUA using QALYs measured in Western countries [8,9,10,11]. A pilot study showed that it is feasible and acceptable to generate Lebanese preference values with the Arabic version of SF-36, which were comparable to those estimated in the UK [12]. The first EQ-5D-5L value set in the Middle East was recently published in Egypt and was determined from a total actual sample of 974 participants, majority at younger age, from both rural and urban areas [13]. In the absence of a country-specific value set, this pilot study is the first to measure HRQoL in Lebanese patients with certain medical conditions using EQ-5D-5L instrument, and to determine demographic and medical factors affecting such health utility scores.

## 2. Material and Method

This was a prospective cross-sectional pilot study in six community pharmacies in the Beirut region from October 2018 through January 2019. Data collection was performed by fifth year student pharmacists, enrolled at the Lebanese American University School of pharmacy, assigned to a community pharmacy site as part of their required pharmacy practice experiences. Every two students were trained on how to fill out the survey and were supervised by one school preceptor and by the site preceptor, at each of the six practice sites. For a period of 3 months, students were on site 4 days a week. For 4 h per day, they approached around 160 potential participants who were visiting the pharmacy, to initiate filling out the survey questionnaire.

The study population consisted of male and female patients older than 18 years, visiting their community pharmacy to collect their prescribed medications for a common condition, cardiovascular, or diabetes mellitus. In total, 102 participants agreed to give informed consent before anonymously filling the predesigned survey, which included questions that mostly capture health related quality of life, and information on health conditions and satisfaction with treatment. The first part of the survey was to collect demographic data, socioeconomic status, and medical management such as duration of drug therapy, number of doctor’s visits, and number of hospitalizations. Participants were asked to rank their overall health condition: before the diagnosis of their current medical condition; with their current medical condition; and with their current medical treatment using a ranking from 1 to 10: one being the worst and 10 being the best status.

The second part of the survey included the questions of the EQ-5D-5L standardized instrument for measuring generic health status and health-related quality of life. Questions of the EQ-5D-5L validated tool address the physical and emotional domains, aiming to reveal important insight into the patient experience with a disease and its treatment. The EQ-5D-5L consists of questions regarding five dimensions: mobility, self-care, usual activities (two questions), pain/discomfort (two questions), and anxiety/depression. For each question, five level categories are possible: no problems, slight problems, moderate problems, severe problems, or extreme problems. The combination of the five dimensions and their five levels results in a health state, each state can be described by a five-digit number that ranges from 11,111 (no problem in any dimensions) to 55,555 (extreme problems in all dimensions) using the UK tariff as well as the recently published Egypt Tariff [13,14]. Outcomes for the EQ-5D-5L could only be calculated for completed surveys. 

The sample size was calculated for medium effect size d = 0.56 [15], a two-tailed type I error of 5%, and a desired power of 80%. Due to limited resources, convenient sampling was used to allow attaining a maximum of 102 participants voluntarily answering the questionnaire.

Statistical methods: SPSS v26 was used to conduct the statistical analysis once data were coded and entered. Sample socioeconomic characteristics, health conditions, and medication use were summarized using frequency and percentage, whereas satisfaction with treatment and ranking of current health were summarized using mean and standard deviation. EQ-5D-5L dimensions were presented as both percentages and means with standard deviations as well as medians with IQR. Differences in EQ-5D-5L mean score among the sample characteristics were tested using either the independent t-test or the analysis of variance with Bonferroni correction for pair-wise comparisons. Differences in median were tested using the non-parametric tests: Mann–Whitney or Wilcoxon. Effect of individual diagnosis was tested without correction for potential effect of multiple testing. Multivariable linear regression model was also used to test for the effect of covariates. All analyses were conducted at the 0.05 significance level. All variables with a *p*-value ≤ 0.2 were entered in the linear regression using, only those with *p*-value < 0.05 in the model were kept. This study was approved by the Lebanese American University Institutional Review Board.

## 3. Results

Out of 102 respondents, 54.5% were males and 45.5% were females. Among the total respondents, 82.4% were above 50 years of age and 77.8% of the respondents were living with their families. Most of the participants classified themselves as either middle-class (39.4%) or high-class (37.4%). In total, 65.7% had been receiving treatment for over a year, 44% made none or one visit to their physician within the previous year, and 67.6% have not been hospitalized in the past 12 months. About two-thirds of participants were diagnosed with one medical condition and, 16.7% had three or more medical conditions; the most common was hypertension (37.3%) followed by diabetes (32.4%), dyslipidemia (26.5%), and cardiovascular diseases (CVD) (16.7%). 

When asked to rank their current health status over a 10-point scale, participants gave an average of 6.8 and 7.4 for their current health out of 10 and for their satisfaction with their treatment, respectively (Table 1). 

In total, 42% of respondents received monotherapy with 31.4% of them were found to be on antidiabetic medications (majority on biguanides), and 35.3% on lipid-lowering medications (majority on statins). Beta-blockers, calcium channel blockers, angiotensin converting enzyme inhibitors/angiotensin receptor blockers were reported to be taken by 25% of respondents (Table 2).

The frequencies of item responses for each EQ-5D-5L dimension are presented in Table 3. The mean utility score was 0.762 (SD 0.202) using the UK value scores, and 0.698 (SD 0.297) using the Egypt value scores with a correlation coefficient R = 0.967. Moderate to incapacitating problems were reported by 20.5% of participants for mobility, by 10.9% for self-care, and by 20.6% for usual activities. Moderate-to-extreme pain/discomfort was reported by 36.3% of participants, and moderate-to-extreme depression/anxiety was reported by 23.6%. 

EQ-5D-5L scores by socio-demographic variables, age and gender are summarized in Table 4. The bivariate analysis shows that men had significantly higher utility scores than women (mean 0.815 (SD 0.148) versus 0.697 (SD 0.240), *p* = 0.003). There were statistically significant differences in scores in terms of age groups (lowest score for ages of 75 and above *p* < 0.001), socioeconomic status (lowest score for those self-identified as low SES, *p* = 0.002), doctors visit in the past year (lower score for those reporting 4 or more visits, *p* = 0.014) and hospital admissions over the past year (lower score for those reporting 2 or more admissions, *p* < 0.001), but not for the living status nor the duration of current treatment.

As presented in Table 5, being diagnosed with hypertension and CVD along with the number of medical diagnoses were shown to have a statistically significant lower score on the EQ-5D-5L (*p* = 0.019, *p* = 0.017, and *p* < 0.001, respectively), although the differences in median was borderline significant for hypertension (*p* = 0.052) and not significant for number of diagnoses (*p* = 0.150). Concerning drug class, only the class of diuretics was shown to have a significant difference in EQ-5D-5L scores with a *p*-value < 0.001. The increased number of prescribed medications per respondent indicated a significant negative impact on the quality of life (*p* = 0.002).

EQ-5D-5L scores were positively correlated with the participants’ ranking of current health status (r = 0.639, *p* < 0.001) as well as with their satisfaction with their current treatment (r = 0.465, *p* < 0.001). Variables were simultaneously tested for their independent association with EQ-5D-5L score using a multivariate linear regression model. Older age, hospitalization in the past 12 months, chronic hypertension, and using 7 or more medications, were all independent predictors of a lower EQ-5D-5L score, whereas a higher score on treatment satisfaction was associated with a higher EQ-5D-5L score. Patients 75 years or older, had on average 0.15 lower score on quality of life (*p* < 0.001). Patients who were hospitalized in the past 12 months had 0.111 lower score on average (*p* = 0.001), while those diagnosed with hypertension had an average 0.065 lower score (*p* = 0.035). Participants reporting the use of 7 or more medications had an EQ score that was 0.282 lower than their counter parts (*p* < 0.001). Finally, for every unit increase in treatment satisfaction the quality-of-life score increased by 0.036 unit (*p* = 0.001) (Table 6).

## 4. Discussion

To our knowledge, this is one of the first studies to use EQ-5D-5L value set to assess the patients’ HRQoL with a common medical condition in Lebanon. High levels of EQ-5D-5L scores are significantly associated with male gender, high socioeconomic status, and frequent doctor’s visits. Moreover, diabetes mellitus or CVD appear to be associated with a lower quality of life similarly to diuretic intake and an increased number of comorbidities. Treatment satisfaction affected positively the quality of life of patients with certain medical conditions whereas older age, hospitalization in the past 12 months, having hypertension, and taking more than seven medications were negative predictors of EQ-5D-5L score.

This study demonstrated an inverse correlation between HRQoL and the number of comorbidities. This inverse relationship was observed elsewhere in various chronic conditions such as dementia, psoriasis, and cancer [16,17,18,19]. Furthermore, the number of prescribed medications equal or greater than seven was negatively correlated with the HRQoL which is in line with previous findings in the literature [20]. In fact, the number of prescribed drugs may be considered as a proxy for general morbidity and another indicator of comorbid conditions that were not actively sought by patients [20]. The HRQoL was affected by the sociodemographic characteristics. Female patients had poorer HRQoL compared to male patients. Many studies demonstrated gender differences in HRQoL among patients with common diseases such as coronary artery disease; for instance, Yinko et al. found that after adjusting for disease characteristics and management, several factors were found to be significantly associated with HRQoL including femininity score, household responsibility and social support [21]. 

Elderly people were more represented in this sample due to the nature of the studied diseases. Increased age was associated with poorer HRQoL in both bivariate and multivariate linear regression. The coefficient for age was greater than that of the frequency of hospitalization and the presence of hypertension. These results should be interpreted with caution as a cut-off of 75 years of age was considered for analysis meaning that only advanced age (>75) is strongly and significantly correlated with poorer health-related quality of life.

More patients lived with their family than alone, owing to the Middle Eastern culture and social norms. The fact that pharmacies are located in urban areas explains why most of the patients reported middle-to-high socioeconomic status. Patients with higher socioeconomic status had statistically significantly better HRQoL. Previous studies demonstrated similar association between level of income, education, social class, and HRQoL [22,23]. More than half of the patients were on their current treatment for more than 12 months, which could be considered a validation of the targeted population. The majority of patients did more than two physician consultations per year, which would reflect the rate of clinic visits for patients with certain chronic conditions, their access to healthcare professionals, and reported socioeconomic status [22,23]. 

The frequency of physician and or hospital visits could be considered as surrogate markers of the severity of the disease and comorbidities, which explains the positive correlation between these two factors and the HRQoL. The study results confirm previous findings in various patient groups showing a link between healthcare resource utilization and HRQoL. EQ-5D-5L was found to be an accurate measure that predicts mortality, emergency department utilization and hospital discharge rates [24]. 

The positive correlation of the treatment satisfaction with HRQoL, further consolidates the results of a study done in Lebanon by Khabaz et al. that showed positive associated between increased adherence to treatment, a higher global satisfaction and an increase in quality of life [25]. While Zhang Li et al. reported that hypertension was related to lower scores in mobility, self-care and usual activity; our findings showed that self-care was the least affected dimension with 72.5% of the people reporting having no problems with self-care. This might be partly explained by the fact that the survey was conducted at community pharmacies where patients were filling their prescriptions in person [26,27]. Similar to a previous study, the score in the domain of pain/discomfort among individuals with hypertension, diabetes or cardiovascular diseases was the most affected dimension [28]. In fact, Zang li et al. reported that hypertensive individuals with body pain/discomfort might have a poorer HRQoL than the general population [27]. Furthermore, this study findings are aligned with a recently published study highlighting the negative impact of diabetes mellitus on the patients’ quality of life in Middle Eastern countries [5]. Additionally, in our study, patients with a history of CVD and hypertension had a negative impact on their quality of life compared to patients without heart disease or hypertension, and this is similar to previous findings [26,27,29]. This negative correlation with the HRQoL was not statistically significant, and was probably due to a low number of participants. In contrast, the evaluation of EQ-5D-5L in China for different chronic diseases including heart diseases, diabetes and hypertension demonstrated a stronger negative correlation of HRQoL with hypertension than with diabetes mellitus [29]. 

The study includes the following limitations: the health utility index could be influenced by the choice of value set used to convert self-classification scores. Selection bias is another limitation due to convenient sampling and the fact that patient recruitment was limited to older age groups with cardiovascular disorders from one geographic area (capital city) and may have limited the subgroup analysis of EQ-5D-5L per disease. Sampling bias may exist because only patients who presented personally to the pharmacy were included. So generally, they might have higher mobility and thus a better quality of life. In the absence of a health utility index specific to Lebanon, EQ-5D-5L UK value set was used as well as the Egypt value set, which has been recently published. A comparison between the use of both value sets in this study, showed a high correlation between the two value sets used on our data. While the health preference values from the UK were relatively comparable to the estimated health preference values in Lebanon using the Arabic version of SF-6D to generate utility values from the SF-36, these values may have limited generalizability, due to the fact that the small sample size may not be representative of the population [12]. While the first value set for EQ-5D-5L in Egypt might be attractive to be used in Lebanon as a country in the MENA region, it might not be the best option for this study targeting a population of older age with CVD or diabetes mellitus, living in an urban area. The Egypt value set was determined from a total actual sample of 974 participants, with a majority of a younger age, from both rural and urban areas [13]. 

The successful application of such an instrument in our population could pave the way to large-scale studies aiming to further evaluate EQ-5D-5L scores and factors affecting such scores in patients with certain conditions. Until a country-specific value set is determined, future studies could also investigate the feasibility to apply the value set of other countries on a larger population size with different medical conditions in Lebanon.

## 5. Conclusions

This pilot study measured health utility scores and factors influencing HRQoL in the Lebanese population. The increase in the number of medications was the factor that most negatively affected the health utility scores, and consequently, the quality of life of our population. Determining population-specific health utility index would help performing cost effectiveness analysis, which would assist policy makers in their decisions process.

## Figures and Tables

**Table 1 ijerph-18-08817-t001:** Demographic characteristics of participants.

	*n*	%
Gender *		
Male	55	54.5%
Female	46	45.5%
Age		
34 and below	5	4.9%
35–49	13	12.7%
50–64	38	37.3%
65–74	25	24.5%
75+	21	20.6%
Living status		
Alone	22	22.2%
With Family	77	77.8%
Socio-economic status		
3–5 Low	23	23.2%
6–7 Middle	39	39.4%
8–10 High	37	37.4%
Current treatment duration		
<3 months	9	8.8%
3–6 months	8	7.8%
7–12 months	18	17.6%
>12 months	67	65.7%
Visits to the doctors in past 12 months		
0–1	44	44%
2–3	38	38%
4 and more	18	18%
Admissions to hospital in past 12 months		
None	69	67.6%
Once	13	12.7%
Two and More	20	19.6%
Medical conditions		
Hypertension	38	37.3%
Diabetes	33	32.4%
Dyslipidemia	27	26.5%
Cardiovascular	17	16.7%
Others	37	36.3%
Number of diagnosis		
1	67	65.7%
2	18	17.6%
3 and more	17	16.7%

* 1 missing for gender, 3 missing for living status, 3 missing for socioeconomic status, 2 missing for visits to the doctors in past 12 months.

**Table 2 ijerph-18-08817-t002:** Type and prevalence of medications used by participants.

	*n*	%
Antidiabetics	32	31.4%
Biguanide	27	26.5%
Other antidiabetic *	23	22.5%
Antidyslipidemic	36	35.3%
Statin	35	34.3%
Other antidyslipidemic *	6	5.9%
Beta blocker	28	27.5%
ARB/ACEI *	25	24.5%
ARB	16	15.7%
ACEI	9	8.8%
Calcium channel blocker	24	23.5%
Antiplatelet/anticoagulant	18	17.6%
Antiplatelet	16	15.7%
Anticoagulant	3	2.9%
Diuretic	15	14.7%
Proton pump inhibitor	10	9.8%
Psychotropics	10	9.8%
Others	31	30.4%
Number of medications		
1	43	42.2%
2–3	30	29.4%
4–6	21	20.6%
7+	8	7.8%

* Other Antidiabetics include DPP4 = dipeptidyl peptidase 4 inhibitors, SGLT2 = sodium glucose cotransporter 2 inhibitors, GLP1 = glucagon-like peptide−1 agonists, Sulfonylurea, Insulin. Other Anti-dyslipidemics = fibrates. Psychotropics include antipsychotic tricyclics, benzodiazepines, serotonins. ACEI = angiotensin-converting enzyme inhibitor. ARB = angiotensin receptor blocker.

**Table 3 ijerph-18-08817-t003:** Descriptive statistics for EQ-5D-5L items and scores.

	*n*	%	Mean Score (SD) Median (IQR) Using UK Tariff	Mean Score (SD) Using Egypt Tariff
Mobility			0.048 (0.058) 0.058 (0.060)	0.090 (0.130)
I have no problems	41	40.2%		
I have slight problems	40	39.2%		
I have moderate problems	15	14.7%		
I have severe problems	3	2.9%		
I am unable to move	3	2.9%		
Self-care			0.021 (0.042) 0.000 (0.050)	0.026 (0.058)
I have no problems	74	72.5%		
I have slight problems	17	16.7%		
I have moderate problems	7	6.9%		
I have severe problems	2	2.0%		
I am unable to take care of myself	2	2.0%		
Usual activities			0.033 (0.040) 0.050 (0.050)	0.038 (0.056)
I have no problems	50	49.0%		
I have slight problems	31	30.4%		
I have moderate problems	16	15.7%		
I have severe problems	5	4.9%		
Pain/Discomfort			0.070 (0.071) 0.063 (0.080)	0.070 (0.779)
I have no pain or discomfort	27	26.5%		
I have slight pain or discomfort	38	37.3%		
I have moderate pain or discomfort	29	28.4%		
I have severe pain or discomfort	7	6.9%		
I have extreme pain or discomfort	1	1.0%		
Anxiety/Depression			0.066 (0.073) 0.078 (0.080)	0.075 (0.102)
I am not anxious or depressed	40	39.2%		
I am slightly anxious or depressed	38	37.3%		
I am moderately anxious or depressed	17	16.7%		
I am severely anxious or depressed	5	4.9%		
I am extremely anxious or depressed	2	2.0%		
EQ-5D-5L score			0.762 (0.202) 0.809 (0.210)	0.698 (0.297)

**Table 4 ijerph-18-08817-t004:** EQ-5D-5L scores by sociodemographic characteristics.

	Mean	SD	*p*-Value	Median	IQR	*p*-Value
Gender						
Male	0.815	0.148		0.859	0.18	
Female	0.697	0.240	0.003	0.738	0.26	0.005
Age groups						
Below 50 ^a^	0.828	0.132		0.819	0.217	
50–64 ^a^	0.804	0.155		0.829	0.205	
65–74 ^a^	0.796	0.130		0.809	0.136	
75+ ^b^	0.590	0.295	<0.001	0.611	0.451	0.008
Living status						
Alone	0.797	0.144		0.819	0.202	
With Family	0.748	0.218	0.328	0.801	0.216	0.602
Socioeconomic status						
3–5 Low ^a^	0.631	0.222		0.680	0.200	
6–7 Middle ^b^	0.796	0.174		0.829	0.145	
8–10 High ^b^	0.799	0.194	0.002	0.829	0.225	<0.001
Duration of current treatment						
<3 months	0.729	0.121		0.738	0.097	
3–6 months	0.739	0.234		0.802	0.380	
7–12 months	0.746	0.180		0.764	0.197	
>12 months	0.774	0.215	0.880	0.829	0.207	0.399
Visits to the doctors in past 12 months						
0–1 ^a^	0.823	0.145		0.863	0.174	
2–3 ^a,b^	0.736	0.209		0.795	0.247	
4 and more ^b^	0.668	0.271	0.014	0.708	0.190	0.014
Hospital admissions in past 12 months						
None ^a^	0.824	0.138		0.859	0.171	
Once ^b^	0.670	0.266		0.730	0.250	
Two and More ^b^	0.610	0.245	<0.001	0.690	0.286	<0.001

^a,b^ Groups with same superscripts letters are not statistically different at the 0.05 level using Bonferroni.

**Table 5 ijerph-18-08817-t005:** EQ-5D-5L score by medical conditions and medication.

	Mean	SD	*p*-Value	Median	IQR	*p*-Value
Medical condition						
Hypertension						
Yes	0.702	0.250		0.746	0.262	
No	0.798	0.158	0.019	0.829	0.197	0.052
Diabetes						
Yes	0.753	0.270		0.861	0.207	
No	0.767	0.162	0.753	0.788	0.18	0.394
Dyslipidemia						
Yes	0.811	0.196		0.864	0.134	
No	0.745	0.203	0.144	0.770	0.199	0.054
Cardiovascular						
Yes	0.657	0.290		0.733	0.198	
No	0.783	0.174	0.017	0.829	0.218	0.038
Number of conditions:						
1 ^a^	0.777	0.183		0.829	0.221	
2 ^a^	0.772	0.144		0.780	0.273	
3 ^a^	0.806	0.107		0.812	0.131	
4 or more ^b^	0.333	0.459	<0.001	0.304	0.758	0.150
Medications:						
Antidiabetics						
Yes	0.770	0.256		0.861	0.202	
No	0.759	0.174	0.829	0.782	0.199	0.227
Antidyslipidemics						
Yes	0.775	0.239		0.844	0.214	
No	0.755	0.180	0.629	0.788	0.194	0.262
Beta blocker						
Yes	0.710	0.264		0.801	0.26	
No	0.782	0.171	0.111	0.821	0.218	0.278
ARB/ACEI *						
Yes	0.711	0.227		0.801	0.257	
No	0.779	0.192	0.149	0.812	0.221	0.153
Calcium channel blocker						
Yes	0.747	0.191		0.744	0.176	
No	0.767	0.206	0.677	0.811	0.223	0.465
Antiplatelet/Anticoagulant						
Yes	0.691	0.242		0.725	0.207	
No	0.778	0.190	0.097	0.829	0.214	0.042
Diuretic						
Yes	0.579	0.295		0.699	0.200	
No	0.794	0.164	<0.001	0.829	0.207	0.001
Number of medications:						
1–2 ^a^	0.791	0.179		0.829	0.236	
3–4 ^a^	0.761	0.187		0.812	0.192	
5–6 ^a^	0.785	0.113		0.809	0.156	
7+ ^b^	0.509	0.353	0.002	0.655	0.438	0.030

* Angiotensin-converting enzyme (ACE) inhibitor and an angiotensin receptor blocker (ARB). ^a,b^ superscripts indicate statistical significance between the groups using Bonferroni correction.

**Table 6 ijerph-18-08817-t006:** A multivariate linear regression of EQ-5D-5L with selected independent variables.

Variable	Coefficient Beta	Standard Error	*p*-Value
Age			
Below 75 (ref)	--		
75 and more	−0.153	0.036	<0.001
Hospitalized in the past 12 months			
No (ref)	--		
Yes	−0.111	0.034	0.001
Hypertension			
No (ref)	--		
Yes	−0.065	0.030	0.035
Medications			
1 to 6 (ref)	--		
7 and more	−0.282	0.076	<0.001
Satisfaction with treatment (score)	0.036	0.011	0.001

Adjusted R^2^ = 50.0%.

## Data Availability

Data is contained within the article. Further details are available on request from the corresponding author.

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
