# Peer review of "Health-Related Quality of Life in Patients with Health Conditions in Lebanese Community Setting"

_ijerph, 2021, doi:10.3390/ijerph18168817_

Round 1

Reviewer 1 Report

I have reviewed the paper and revised paper before. In general, the authors did not well respond in their first revision; however, in this round of review, I am relatively satisfied with the paper. Specifically, the authors have reorganized their study purpose and aligned their findings to the purpose. My remained concern is the readability is not that smooth. However, this should be tackled using English editing service.

Author Response

thank you for your feedback. writing was reviewed and edits done using track changes.

Reviewer 2 Report

Of course scores are often analyzed using mean and sd, but that doesn't mean its the correct way. The literature is full of badly conducted studies and wrong statistical analyses, as I am sure you are aware. If your findings don't change when using nonparametric methods, that should give you reassurance about them being correct.

Thus, I recommend you report median (IQR) and the results of the nonparamtric tests in your final manuscript.

Author Response

Thank you for your feedback. as recommended we added to the 3 tables the results of the nonparametric tests, reported median (IQR) and edited the text writing in the method/result sections accordingly (highlighted in purple color).

This manuscript is a resubmission of an earlier submission. The following is a list of the peer review reports and author responses from that submission.

Round 1

Reviewer 1 Report

My main concerns regarding the statistical analyses have not been addressed.

  1. The main outcome HRQoL is measured as a score. Thus, it is an ordinal variable and should be treated as such. In particular, means and SDs are no adequate measures for ordinal data. Instead, median and IQR should be reported. The same concern also applies to the tests and the linear regression model performed. For ordinal data, nonparametric methods are better suited.
  2. It does not become clear how exactly the regression model was constructed. What do the p<= 0.2 refer to? Univariate models? Anyway, this type of data-dependent variable selection in regression models has several undesirable properties, increasing the risk of overfit (ie, modeling data too closely, such that future generalizability is
    reduced) and making many statistics, such as the 95% CI, highly questionable. Ideally, variable selection should be guided by expert knowledge rather then data-dependent.
  3. It is not stated in the statistical methods which kind of correlation coefficient is computed. Due to the ordinal data, Spearman should be used.

Reviewer 2 Report

Although the authors have tried to revise their work, their comments do not persuade me that they have made clear regarding that their study design can fulfill their study purposes. If the authors want to see if the EQ5D-5L can be used in the studied population, they should test the psychometric properties  and other features for the EQ5D-5L. The authors said that "This study successfully measured health utility scores and factors influencing HRQoL in the Lebanese population. Such data would assist policy makers in their reimbursement decisions process." However, anyone can use the EQ5D-5L scores in any population and claimed that the EQ5D-5L can be successfully measured. If the authors want to make such a claim, they should have hypotheses in mind and make sure the the EQ5D-5L can satisfactorily address all the hypotheses. Therefore, I cannot see why the present study can make such a claim.